# EphA7 promotes myogenic differentiation via cell-cell contact

**Laura L Arnold[1†‡], Alessandra Cecchini[1,2†], Danny A Stark[1§], Jacqueline Ihnat[1], Rebecca N Craigg[1], Amory Carter[1], Sammy Zino[1#], DDW Cornelison[1,2]\***

[1]Division of Biological Sciences, University of Missouri, Columbia, United States; [2]Christopher S. Bond Life Sciences Center, University of Missouri, Columbia, United States

**Abstract** The conversion of proliferating skeletal muscle precursors (myoblasts) to terminally-differentiated myocytes is a critical step in skeletal muscle development and repair. We show that EphA7, a juxtacrine signaling receptor, is expressed on myocytes during embryonic and fetal myogenesis and on nascent myofibers during muscle regeneration in vivo. In *EphA7*[-/-] mice, hindlimb muscles possess fewer myofibers at birth, and those myofibers are reduced in size and have fewer myonuclei and reduced overall numbers of precursor cells throughout postnatal life. Adult *EphA7*[-/-] mice have reduced numbers of satellite cells and exhibit delayed and protracted muscle regeneration, and satellite cell-derived myogenic cells from *EphA7*[-/-] mice are delayed in their expression of differentiation markers in vitro. Exogenous EphA7 extracellular domain will rescue the null phenotype in vitro, and will also enhance commitment to differentiation in WT cells. We propose a model in which EphA7 expression on differentiated myocytes promotes commitment of adjacent myoblasts to terminal differentiation.

**\*For correspondence:**
cornelisond@missouri.edu

[†]These authors contributed equally to this work

**Present address:** [‡]Medline Industries Inc, Northfield, United States; [§]Ceva Animal Health, LLC, Lenexa, United States; [#]Department of Gastroenterology, Washington University Medical School in St. Louis, St. Louis, United States

**Competing interests:** The authors declare that no competing interests exist.

## Introduction

Skeletal muscle cells (myofibers) are large, syncytial cells which can span the entire length of a limb segment: in humans, the sartorius muscle can be ~60 cm long, with individual muscle fibers longer than 20 cm (*Harris et al., 2005*). Myofibers are generated by the fusion of terminally postmitotic myocytes, which differentiate from proliferation-competent myoblasts. Due to the linear, one-way succession of proliferating myoblast to differentiated myocyte to syncytial myofiber, transitions between states are tightly regulated: either failure to progress from myoblast to myocyte or precocious differentiation from myoblast to myocyte will lead to a deficit of functional contractile muscle. Because of the syncytial nature of myofibers, in skeletal muscle there exists an additional aspect of the decision to commit to differentiation: terminally-differentiated myocytes must have a sufficient number of other fusion-competent myocytes in close proximity to fuse with, or they cannot generate a functional myofiber.

It is a common observation that sparse plating of myogenic cells in vitro delays myogenic differentiation, while cells cultured at higher confluence exhibit a much higher degree of differentiation regardless of pro-mitogenic conditions such as high serum. This has been described as a version of the 'community effect', a phenomenon first noted by John Gurdon in the context of amphibian muscle development (*Gurdon, 1988*). He found that single mesoderm cells or aggregates of less than 100 mesoderm cells will not express MyoD and differentiate into muscle even under conditions that promote myogenesis, while aggregates of 100 or more cells would differentiate efficiently (*Gurdon et al., 1993*); later experiments showed that the homotypic cell-cell adhesion molecule N-cadherin is responsible for at least a portion of this effect (*Holt et al., 1994*). Similar studies in mouse suggested that a minimum of 30–40 cells is required for myogenic differentiation (*Cossu et al., 1995*). As noted earlier, skeletal muscle fibers are syncytial cells formed following

permanent withdrawal of myogenic precursors from the cell cycle: it would make sense that before committing to such a course of action, a potential myocyte would like some assurances that if it 'takes the plunge', other differentiated cells would be available for fusion. Similarly, it seems practical for a signal conveying this information to be contact-mediated.

Ephs are a family of receptor tyrosine kinases that act via juxtacrine interactions with cells presenting their ligands (ephrins) to modify cell motility, assortment, proliferation, differentiation, and survival in multiple tissue types (*Klein, 2010*; *Klein, 2012*; *Kania and Klein, 2016*). Here we present data suggesting that EphA7, a member of this family of bidirectional signaling molecules, is a potent mediator of the community effect. EphA7 is expressed during muscle development and regeneration on differentiated myocytes and nascent myofibers; myogenic cells lacking EphA7 exhibit delayed and prolonged differentiation in vitro and in vivo; and exposing myogenic cells (with or without endogenous EphA7) to EphA7 ectodomain accelerates differentiation. We propose a model in which EphA7 expression on differentiated myocytes promotes synergistic differentiation of adjacent myoblasts. Early-expressing cells differentiate stochastically and express EphA7, which then promotes differentiation and EphA7 expression in adjacent cells, which then promote differentiation and EphA7 expression in cells adjacent to them, etc. We propose that juxtacrine signaling via EphA7 provides a mechanism to promote collective differentiation in myogenic populations, to ensure that myoblasts are in close proximity to fusion-competent myocytes before they commit to terminal differentiation.

## Results

### EphA7 is expressed by differentiated myocytes and nascent myofibers in vivo and in vitro

Our initial observations in an expression screen for Eph/ephrins in the context of adult skeletal muscle regeneration suggested that EphA7, while absent during muscle homeostasis, is transiently upregulated by regenerating myofibers in vivo after a barium chloride ($BaCl_2$) injury (*Stark et al., 2011*). This expression pattern raised the possibility that EphA7 might function during satellite cell differentiation and/or fusion during the acute stages of muscle differentiation. To define when and where EphA7 is expressed during regeneration, we stained sections of uninjured adult mouse tibialis anterior (TA) muscles in parallel with sections of TAs at 3, 5, 7, 10, 16, and 21 days post injury (dpi) following $BaCl_2$ injury for EphA7, and costained with laminin to identify myofiber boundaries and embryonic myosin heavy chain (eMyHC) to identify newly-generated myofibers (*Figure 1A*). We noted that EphA7 protein was consistently expressed by nascent myofibers at all timepoints examined, and that as muscle regeneration was resolved, muscle fiber caliber increased, and eMyHC expression was lost, EphA7 expression was also extinguished (*Figure 1B*). Because multiple splice isoforms of EphA7 exist and the antibody used cannot distinguish between them, we performed RT-PCR on injured and uninjured TA muscle to identify potential alternative splice isoforms and to correlate protein expression with mRNA expression. We found that full-length EphA7 is the most abundant isoform expressed during muscle regeneration, although both truncated isoforms were also detected at varying levels. (*Figure 1—figure supplement 1*).

To quantify EphA7 expression in satellite cells and their progeny, we explanted satellite cells either with or without the host myofiber and stained them with anti-EphA7 alone and in conjunction with anti-Pax7 (to mark myogenic progenitors), anti-MyoD (to mark myoblasts/myocytes), or anti-myogenin (to mark differentiated myocytes). When satellite cells were isolated and cultured in association with their host myofiber, expression of EphA7 was first detected on relatively few cells at 48 hours after isolation, but was seen on the majority of satellite cells by 96 hours in culture (*Figure 2A*). In monoculture, EphA7 expression is detectable only in a few dispersed cells when they are initially plated under low serum conditions (*Figure 2B*), but one day later it is present in individual cells or more often groups of cells in close association with one another and/or with bipolar morphology suggesting differentiation. After three days in low serum, the majority of cells are EphA7$^+$. At all timepoints tested we saw that expression of EphA7 and Pax7 are mutually exclusive, while EphA7 and myogenin are uniformly coexpressed, suggesting that EphA7 expression is restricted to committed myocytes (*Figure 2C*). MyoD, which is expressed by both proliferating and differentiating myogenic cells, is present in a subset of EphA7$^+$ cells, presumably differentiated myocytes.

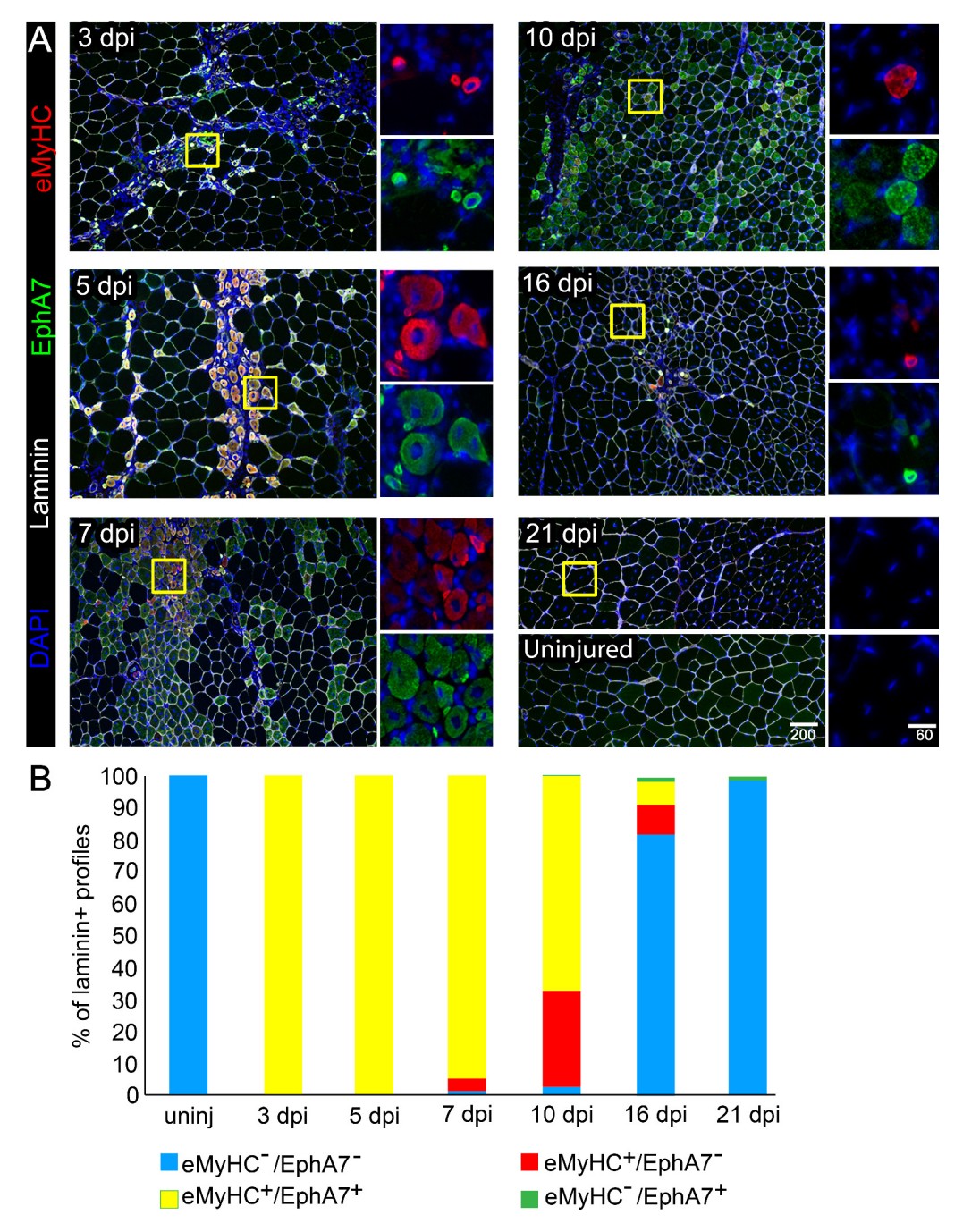

**Figure 1.** EphA7 expression is restricted to nascent myofibers after injury. (**A**) Timecourse of sections of the TA muscle after partial injury via BaCl$_2$ injection. Indirect immunohistochemistry with anti-EphA7 (green), anti-laminin (white) and anti-eMyHC (red) shows that eMyHC expression and EphA7 expression both begin shortly after injury (3 days) and are extinguished when the injury is resolved (by 21 days.) EphA7 staining is limited to laminin$^+$ profiles (nascent or hypertrophying myofibers). T = days post-injury (dpi), boxes indicate regions of color-separated insets. Scale bars = 200 µm in main images, 60 µm in enlarged insets. (**B**) Coexpression of EphA7 and eMyHC over the timecourse of injury. At early timepoints EphA7 and eMyHC are uniformly coexpressed; as the injury resolves (starting at seven dpi) EphA7 and the eMyHC are downregulated. Infrequent fibers (0.3–1.5%) at later timepoints were scored positive for EphA7 but negative for eMyHC based on staining intensity thresholds.

The online version of this article includes the following source data and figure supplement(s) for figure 1:

**Source data 1.** EphA7 and embryonic myosin heavy chain coexpression during regeneration.

**Figure supplement 1.** Expression of splice or developmental isoforms during regeneration.

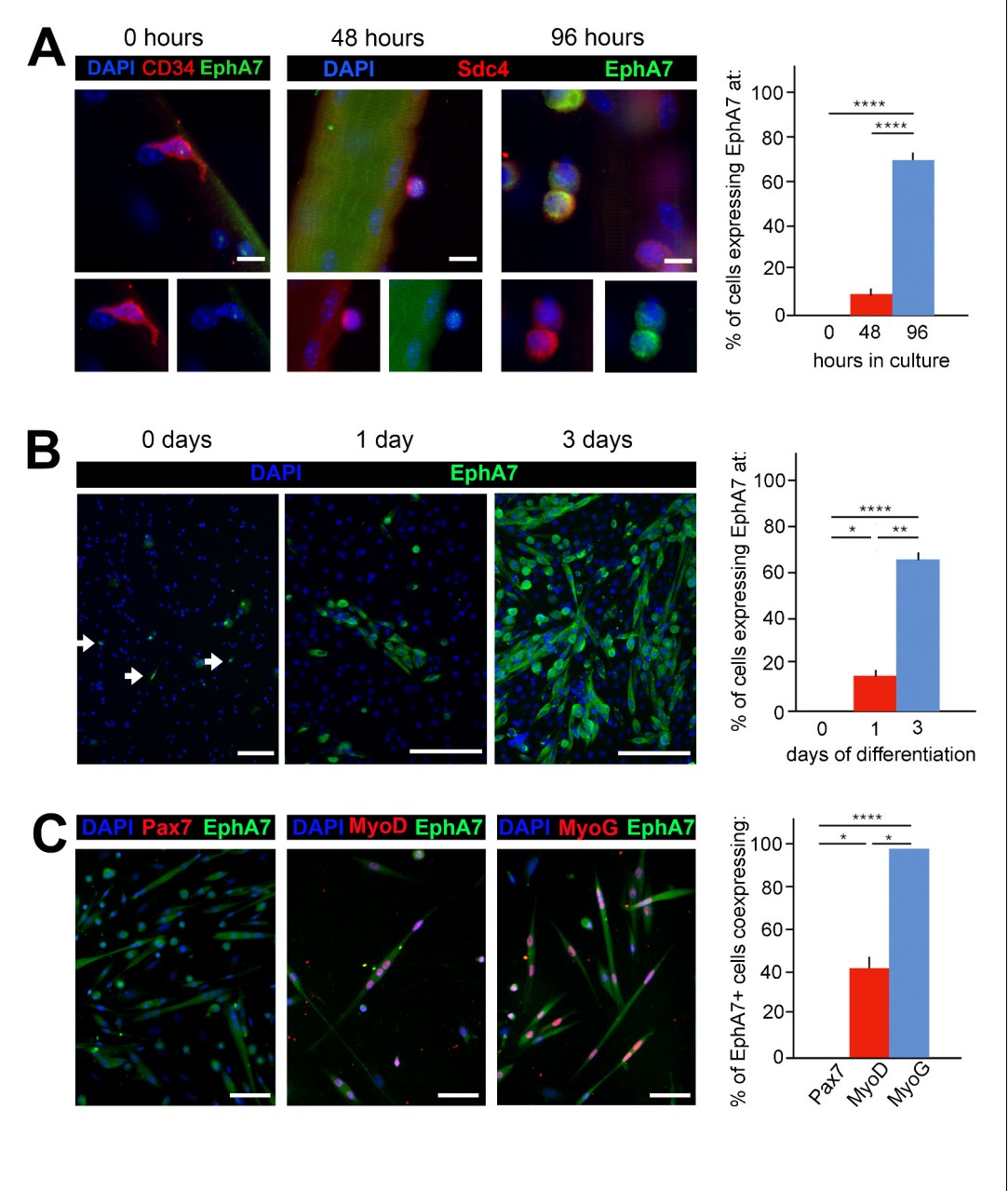

**Figure 2.** EphA7 is upregulated on satellite cells as they commit to differentiation. (**A**) EphA7 expression is absent on satellite cells of freshly-isolated myofibers (marked by expression of CD34 or syndecan-4), but is detectable by 48 hours and prominent by 96 hours. Scale bars = 10 µm. (**B**) Similarly, primary satellite cells in monoculture do not express EphA7 in significant numbers when initially plated in low serum conditions (0 days); rare EphA7+ cells indicated by arrows. However, within 24 hours (1 day) its expression is upregulated under differentiation-promoting conditions; note that during early differentiation EphA7+ cells are present in clusters rather than evenly dispersed. After 3 days in differentiation medium, the majority of cells are expressing EphA7. Scale bars = 100 µm. (**C**) Expression of EphA7 is mutually exclusive with the progenitor cell marker Pax7 but always coincident with the differentiation marker myogenin, confirming specific expression in myocytes (image is of cells after 3 days in low serum). A subset of MyoD+ cells, which would include both myoblasts and myocytes, coexpress EphA7. Scale bars = 100 µm. Error bars = SEM, p ≤ 0.05 (*), 0.01 (**), 0.001 (***) or 0.0001 (****).

The online version of this article includes the following source data for figure 2:

**Source data 1.** EphA7 expression and coexpression in vitro.

To ask whether EphA7 expression is also correlated with myogenic differentiation during development, we repeated the expression studies on sections of developing mouse embryos. Prior work suggests that EphA7 mRNA is present by e11.5 (*Alonso-Martin et al., 2016*), when myogenic differentiation in the somites and limb buds will have already begun. At e11.5, we observed EphA7 protein staining in the dermomyotome on the surface of cells also expressing MyoD (*Figure 3A*), although populations of MyoD$^+$ cells at the dorsomedial lip (which will migrate under the dermomyotome then differentiate to form the primary myotome) and the ventrolateral lip (which will migrate to the limb buds then differentiate to form the limb musculature) do not express EphA7. When costained for myogenin in equivalent sections, EphA7$^+$ cells in the somite all express myogenin, indicating that they have committed to differentiation (*Figure 3B*). This can be more easily noted in oblique sections through the developing somites, in which mononuclear, differentiated primary myotome fibers can be viewed from end to end; in these sections we see that the most recently-differentiated cells express myogenin but not yet EphA7 (arrow). Finally, EphA7 is expressed in all myosin heavy chain (MyHC)-expressing differentiated myocytes in the somite (*Figure 3C*). By e15.5, we observe EphA7 expression on differentiated, myogenin$^+$/MyHC$^+$ myocytes and myofibers in muscles of forelimbs (*Figure 3D*) and trunk (diaphragm and body wall) (*Figure 3E*) as well.

## Muscles from *EphA7*$^{-/-}$ mice are characterized by decreased myofiber size, myofiber number, myonuclear number, and progenitor cell number

To test for a functional role of EphA7 during muscle development and homeostasis, we obtained mice carrying a germline deletion of EphA7. This strain has multiple nonlethal defects in the central nervous system (*Rashid et al., 2005*; *Clifford et al., 2014*; *Kim et al., 2016*) but no skeletal muscle phenotype has been described. We noted that *EphA7*$^{-/-}$ mice are smaller than WT littermates at birth, and hindlimb muscle wet weight is reduced in adult mice (*Figure 4A*). The decrease appears to be due to both a reduction in myofiber number and in myofiber size (*Figure 4B*). In addition, myofibers from *EphA7*$^{-/-}$ mice had fewer myonuclei and fewer myogenic precursor cells (*Figure 4C*) compared to WT.

## Muscle regeneration in *EphA7*$^{-/-}$ mice is protracted and characterized by the persistence of immature myofibers

To test for a functional role of EphA7 during muscle regeneration, we injured TA muscles of WT and *EphA7*$^{-/-}$ mice with the myonecrotic agent barium chloride (BaCl$_2$) (*Caldwell et al., 1990*; *Morton et al., 2019*) and harvested the muscle at 5, 10, 14, and 28 dpi. As noted above, during regeneration nascent myofibers can be identified by their expression of developmental isoforms of myosin heavy chain such as eMyHC; regenerated myofibers can be identified by the presence of centrally located rather than peripheral myonuclei, a condition that persists long after regeneration has been completed (*Meyer, 2018*). While *EphA7*$^{-/-}$ muscles do regenerate, we noted a persistence of eMyHC expression at 10 dpi, by which time eMyHC was downregulated in most regenerated fibers in WT muscles (*Figure 5A*). On closer examination (*Figure 5B–C*), some eMyHC$^+$ myofiber profiles in regenerated *EphA7*$^{-/-}$ muscle could be identified as unfused myocytes or small, mononucleated myofibers lacking a basal lamina. While muscle fiber diameter had returned to baseline by 28 days in both genotypes, hypertrophy was delayed in the *EphA7*$^{-/-}$ muscles both initially (between 5 and 10 dpi) and later when WT fiber diameter had nearly recovered (14 dpi) (*Figure 5D*).

## *EphA7*$^{-/-}$ satellite cells have a delayed and protracted transition from proliferating myoblasts to differentiated myocytes in vitro

One possible explanation for the decreases in myonuclear number, myofiber diameter, and myofiber number observed in *EphA7*$^{-/-}$ muscle would be a proliferation deficit in myogenic precursor cells or myoblasts. An alternative hypothesis would be that the transition from one myogenic specification state to the next is inhibited in the absence of EphA7. To determine whether there is a specific step in the sequence of myogenesis during which EphA7 is required, we assayed markers of activation, proliferation, and differentiation in WT and *EphA7*$^{-/-}$ satellite cells in vitro. We noted that while cell proliferation and upregulation of MyoD were not significantly different between WT and *EphA7*$^{-/-}$

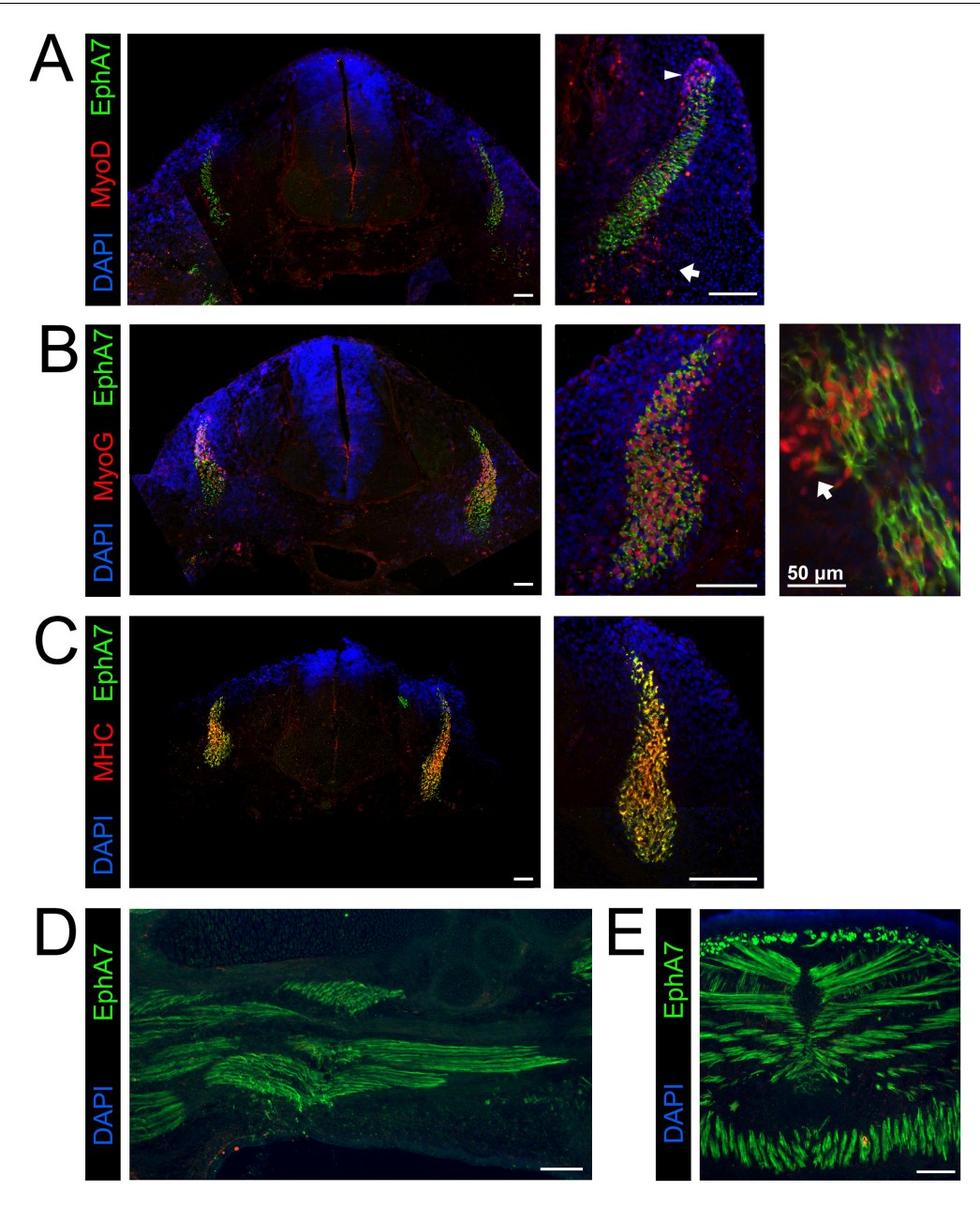

**Figure 3.** EphA7 is expressed on differentiated myogenic cells during development. e11.5 embryos sectioned at the level of the forelimb show expression of EphA7 in differentiated myogenic cells in the somites. (A) While some MyoD-expressing cells in the somite coexpress EphA7, undifferentiated MyoD$^+$ migratory myoblasts at the dorsal-medial lip (right panel, arrowhead) and the ventral-lateral lip (right panel, arrow) do not. (B) Myogenin$^+$ myocytes and primary myofibers in the myotome of the somite are almost all positive for EphA7; in oblique sections (far right panel; scale bar = 50 µm) the most recently-differentiated myogenin$^+$ cells (arrow) do not yet express EphA7. (C) Expression of myosin heavy chain in the somite is uniformly coincident with EphA7 expression. (D, E) In e15.5 embryos, EphA7 also identifies myofibers in the forelimb bud (anterior at top) and diaphragm/body wall muscle. Scale bars = 100 µm unless otherwise noted.

satellite cells cultured on their host myofibers at 48 hours after myofibers (*Figure 6A–B*), upregulation of myogenin expression was greatly decreased in *EphA7$^{-/-}$* satellite cells at 96 hours (*Figure 6C*). Similarly, the fraction of satellite cells in monoculture expressing of Pax7, proliferating (not shown), and expressing MyoD (*Figure 6D*) did not differ between WT and *EphA7$^{-/-}$* cells, but

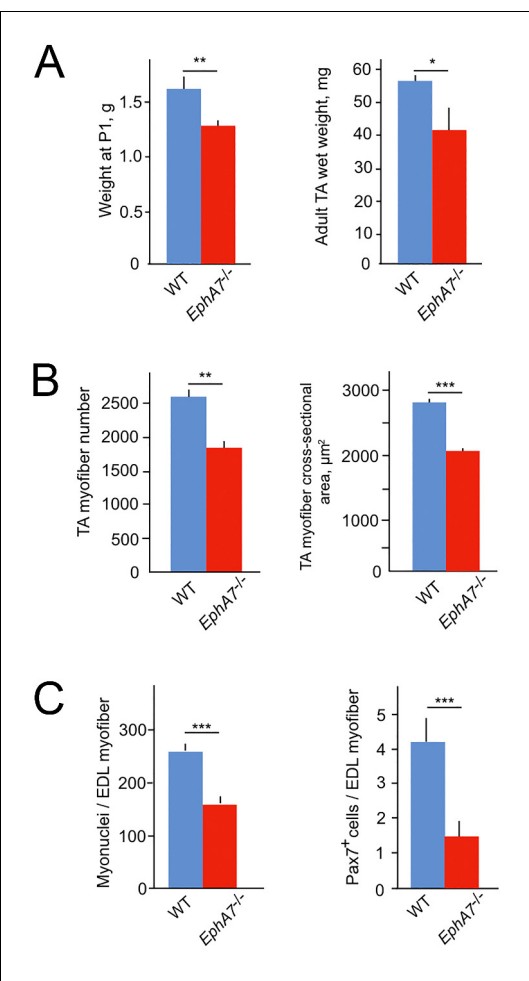

**Figure 4.** *EphA7⁻/⁻* mice have reduced muscle mass caused by diminished myofiber size and number. (**A**) *EphA7⁻/⁻* mice weigh less that WT littermates at birth and continue to have reduced muscle mass as adult animals. (**B**) This is correlated with a reduction in myofiber number as well as myofiber size. (**C**) Individual muscle fibers in *EphA7⁻/⁻* mice possess fewer myonuclei and have reduced numbers of progenitor cells during homeostasis. Error bars = SEM, $p \leq 0.05$ (*), 0.01 (**), or 0.001 (***).

The online version of this article includes the following source data for figure 4:

**Source data 1.** Morphometric analysis of EphA7-/- mice.

expression of myogenin was significantly decreased (*Figure 6E*). *EphA7⁻/⁻* cultures had a higher prevalence of round, birefringent cells (typical of undifferentiated myoblasts) but otherwise there were no notable differences in cell morphology in vitro.

Because the *EphA7⁻/⁻* mouse used is a conventional knockout, it is possible that some of the effects noted during development and regeneration in vivo are due to loss of EphA7 in nonmuscle cell types. Other cell types resident in the muscle have been shown to modify satellite cell-mediated muscle regeneration, particularly fibroadipogenic progenitor cells (FAPs) (*Joe et al., 2010*; *Murphy et al., 2011*). To address this, we queried existing expression datasets for FAPS specifically as well as for all muscle-associated cells. EphA7 is not expressed by FAPS either during homeostasis or injury (based on bulk RNA-seq, *Joe et al., 2010*). Combined with the presence of the phenotype during both development and regeneration, the persistence of the phenotype in muscle cell culture ex vivo, and the recapitulation of the phenotype in myogenic cell lines carrying engineered mutations (below), we propose that expression of EphA7 by myocytes is the primary driver of the in vivo and in vitro phenotype we describe here.

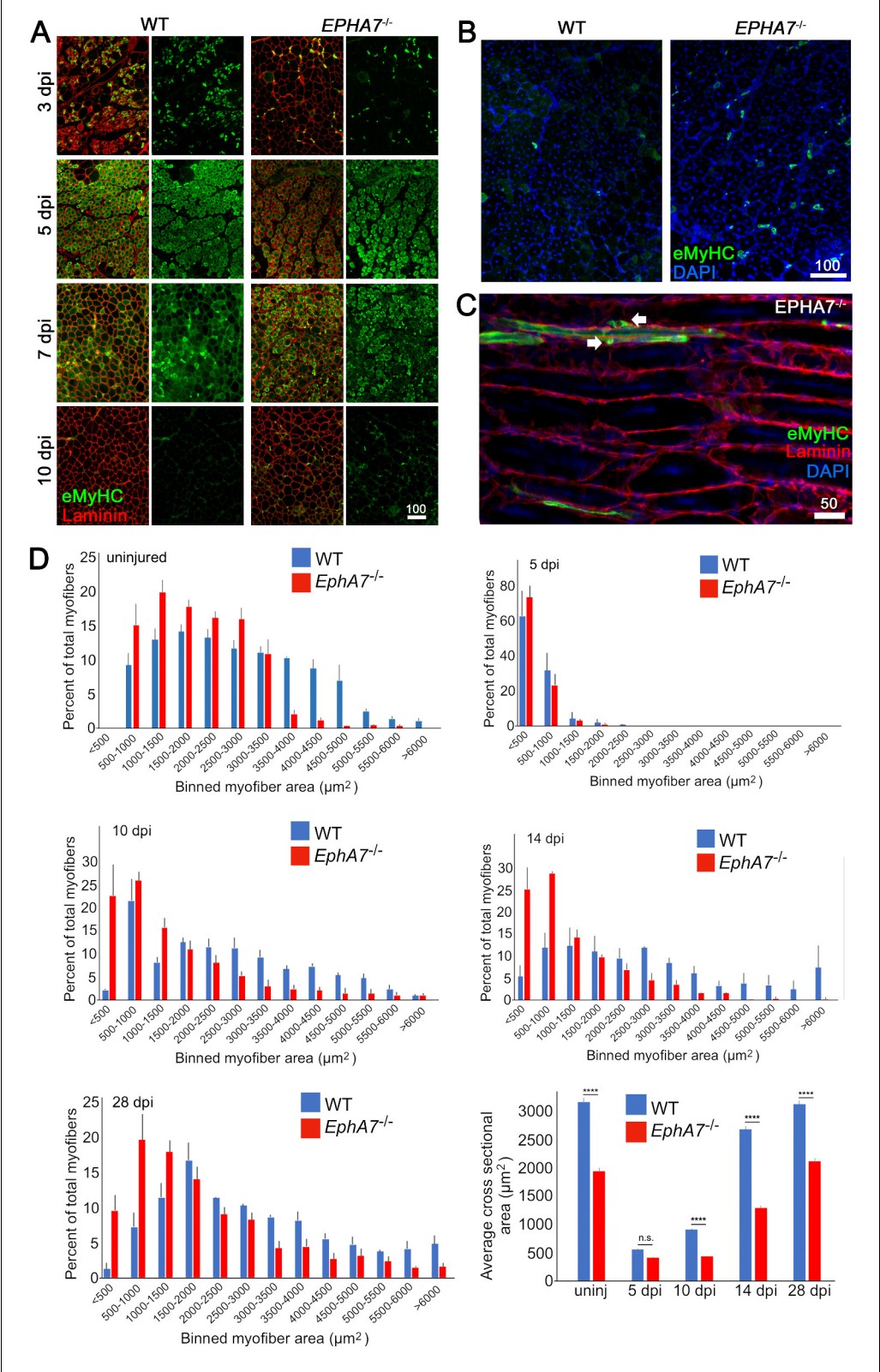

**Figure 5.** *EphA7*[-/-] muscles regenerate after acute injury but maturation and hypertrophy are delayed. (**A**) After acute injury, *EphA7*[-/-] muscles initially possess fewer nascent (embryonic myosin heavy chain[+]) myofibers early in regeneration than WT (3 dpi). At later time points eMyHC expression persists in *EphA7*[-/-] muscles when it has become extinguished in favor of adult isoforms in WT (7–10 dpi). (**B, C**) Closer comparison at 10 dpi shows persistent small-caliber eMyHC+ profiles in cross sections of *EphA7*[-/-] muscles, which in sagittal section are recognizable as either small nascent myofibers with

*Figure 5 continued on next page*

*Figure 5 continued*

few myonuclei or differentiated but unfused myocytes (arrows). We also note eMyHC⁺ cells/processes that do not appear to possess a basal lamina (lower left), another indication of immaturity. Scale bars = 50 or 100 µm as indicated. (D) Quantitation of myofiber caliber in uninjured WT or *EphA7⁻/⁻* muscles confirms the disparities in myofiber size between genotypes, which is eliminated in early stages of regeneration (5 dpi) but is reconstituted at 10, 14 and 28 dpi. Average myofiber area in *EphA7⁻/⁻* muscles was not significantly different between 5 and 10 dpi, or in either genotype between 28 dpi and uninjured muscle. Error bars = SEM, p ≤ 0.0001 (****).

The online version of this article includes the following source data for figure 5:

**Source data 1.** Comparison of regeneration in WT and EphA7-/- muscle.

## Exogenous EphA7 ectodomain induces myogenic differentiation in *EphA7⁻/⁻* and WT satellite cells

While these data suggest that expression of EphA7 promotes myogenic differentiation, they do not distinguish between a cell-autonomous or a non-cell-autonomous mechanism. Eph:ephrin signaling can be either 'forward' (Ephs act as receptors) or 'reverse' (Ephs act as ligands), thus a cell-autonomous mechanism would suggest failure of *EphA7⁻/⁻* cells to receive a signal, while a non-cell-autonomous mechanism would suggest loss of the signal itself. Because EphA7 expression is limited to myocytes and loss of EphA7 appears to affect myoblasts, we hypothesized that in this context EphA7 acts non-cell-autonomously via reverse signaling. If this is correct, then expression of EphA7 should be dispensable provided that exogenous EphA7 is present in the culture; furthermore, only

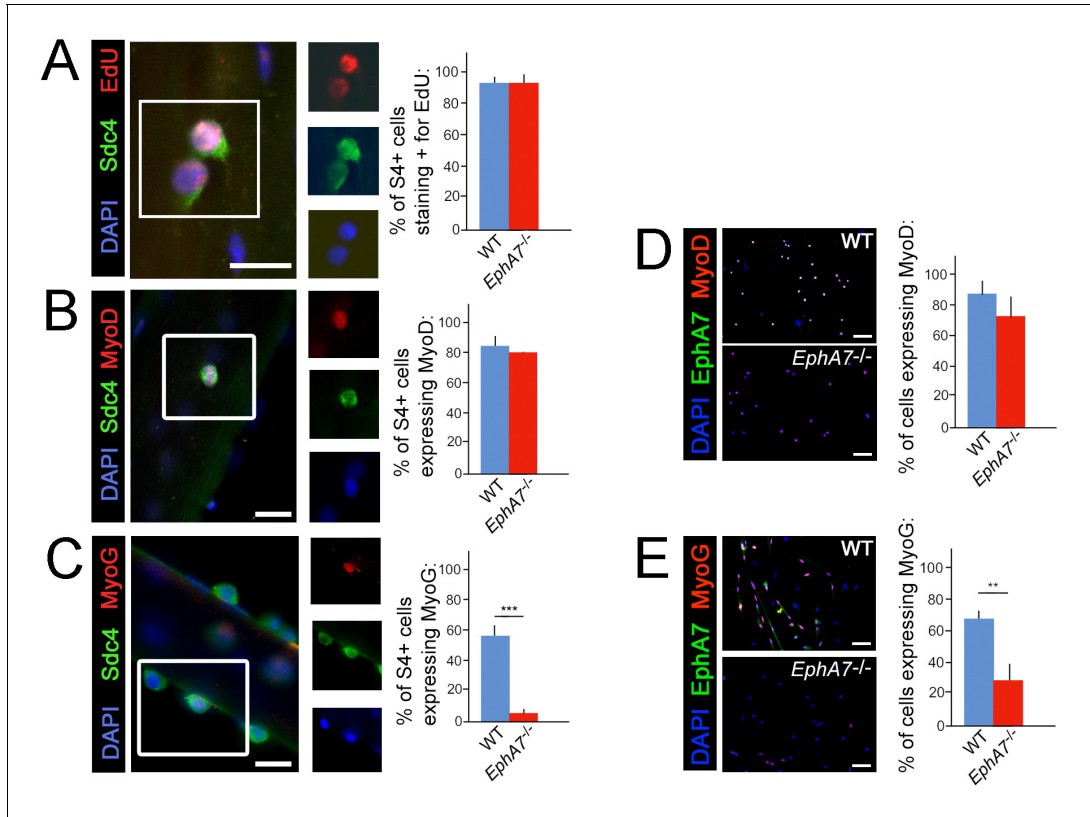

**Figure 6.** Upregulation of terminal differentiation markers is delayed in *EphA7⁻/⁻* myoblasts. (A, B) In single fiber culture, the percentage of syndecan-4⁺ satellite cells incorporating EdU or expressing MyoD at 48 hours is not significantly different between WT and *EphA7⁻/⁻*. (C) However, *EphA7⁻/⁻* satellite cells largely fail to upregulate myogenin (MyoG) at 96 hr. (D, E) Similarly, WT and *EphA7⁻/⁻* satellite cells grown in monoculture express MyoD at statistically similar levels 48 hours after plating, but *EphA7⁻/⁻* cells do not upregulate myogenin to the same extent as WT 96 hours after plating. Scale bars = 10 µm. Error bars = SEM, p ≤0.01 (**) or 0.001 (***).

The online version of this article includes the following source data for figure 6:

**Source data 1.** Coexpression of EphA7 with markers of proliferation, specification, and differentiaion.

the ectodomain (rather than the full-length protein) should be sufficient for this effect. We would therefore predict that exposure to EphA7 ectodomain would rescue the differentiation delay observed in *EphA7⁻/⁻* cells and enhance differentiation of WT cells by acting as a proxy for high cell density. To test this, we plated WT and *EphA7⁻/⁻* cells at equivalent densities on either laminin-coated coverslips or laminin-coated coverslips functionalized with a chimeric EphA7 ectodomain fused to human Fc (EphA7-Fc) (R&D Systems). Every 12 hours, we assayed the fraction of cells that had differentiated (as measured by expression of myogenin). We found that even at the earliest time assayed (12 hours after plating), the fraction of *EphA7⁻/⁻* cells on EphA7-Fc scoring positive for myogenin was equivalent to that of WT cells on laminin, while WT cells on EphA7-Fc differentiated to a greater extent than WT cells on laminin alone (*Figure 7A*). These results were consistent over the course of the assay, and by 48 hours of exposure *EphA7⁻/⁻* cells stimulated with EphA7-Fc showed myogenin expression and differentiated morphologies indistinguishable from WT (*Figure 7B*). We therefore conclude that EphA7 acts non-cell-autonomously as a myocyte-derived signal to promote rapid, synergistic terminal differentiation in local myoblasts, consistent with observations of the community effect.

## Ephrin-A5 expressed on myoblasts mediates the effects of EphA7

If EphA7 is signaling in reverse to myoblasts in order to promote their differentiation, we would expect there to be an ephrin (most likely an ephrin-A) present on myoblasts which can transduce the EphA7 signal to promote myogenic differentiation. In multiple contexts in vivo, EphA7 signals in reverse through ephrin-A5 (*Holmberg et al., 2000*; *Miller et al., 2006*), and in an expression screen for all mammalian ephrins on WT and *EphA7⁻/⁻* satellite cells we noted that ephrin-A5 is present on cells of both strains but shows increased expression in *EphA7⁻/⁻* cells (*Figure 8A*). This is consistent with upregulation of ephrin-A5 in other contexts in the *EphA7⁻/⁻* mouse (*Miller et al., 2006*). If ephrin-A5 is the EphA7 receptor, we would expect that cells lacking ephrin-A5 would phenocopy the *EphA7⁻/⁻* cells' differentiation defect, but that unlike *EphA7⁻/⁻* cells they should not be rescued by exogenous EphA7 ectodomain. To test this, we generated ephrin-A5ᴷᴼ C2C12 cells using a double-nickase strategy and isolated a clonal population carrying an inactivating deletion, as well as a similar clonal population of EphA7ᴷᴼ C2C12 cells (*Figure 8B*); we refer to these mutant cell lines as 'KO' to avoid confusion with mutant cells from the null mouse (*EphA7⁻/⁻*). Importantly, EphA7ᴷᴼ C2C12 cells

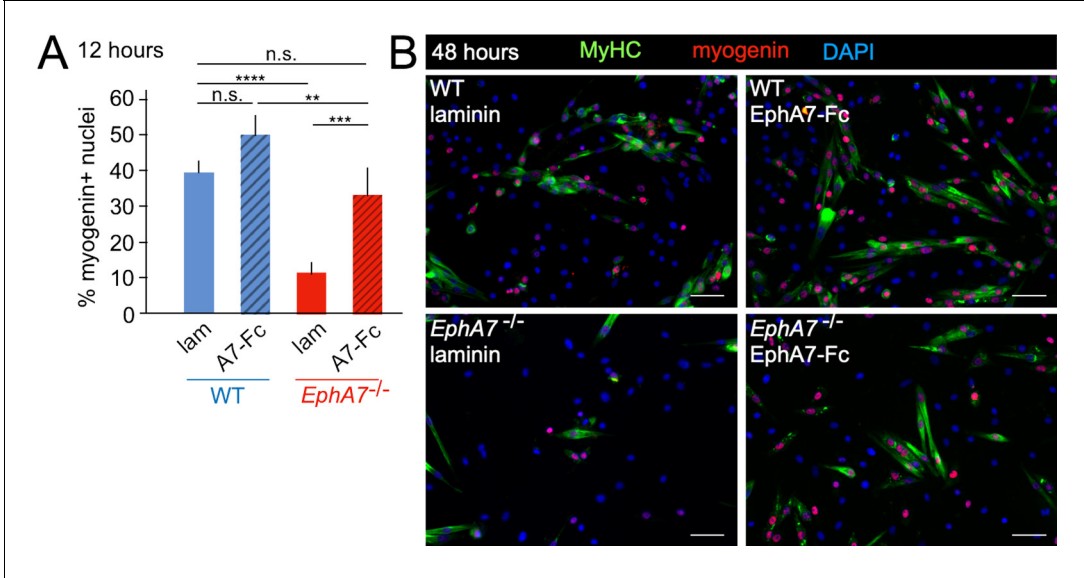

**Figure 7.** Exogenous EphA7 extracellular domain rescues differentiation of *EphA7-/-* satellite cells. WT and *EphA7⁻/⁻* cells were cultured on coverslips coated with either laminin alone (lam) or laminin functionalized with EphA7-Fc chimera (A7–Fc) in growth medium (F-12 + 15% horse serum). (**A**) 12 hours of exposure to EphA7-Fc rescued differentiation of *EphA7⁻/⁻* cells (scored by expression of myogenin) to WT levels, while more WT cells on EphA7-Fc were differentiated than on laminin. Error bars = SEM, p ≤ 0.01 (**), 0.001 (***) or 0.0001 (****). (**B**) *EphA7⁻/⁻* myoblasts acquire morphological and molecular characteristics of differentiated myocytes similar to WT cells when exposed to EphA7-Fc. Scale bars = 50 µm.

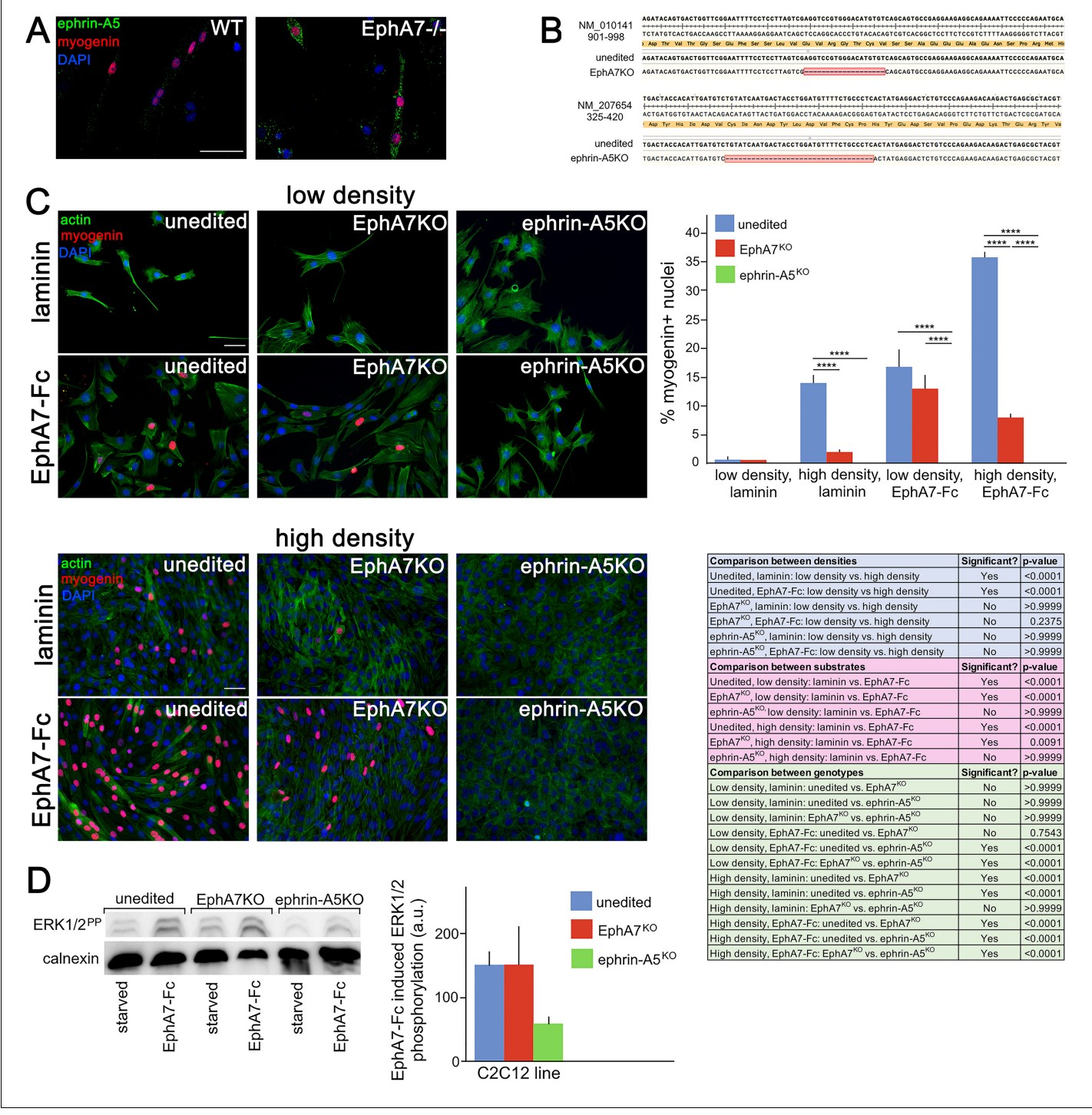

**Figure 8.** Ephrin-A5 meets the criteria for the EphA7 receptor on myoblasts. (**A**) Ephrin-A5 is expressed at higher levels on *EphA7*[-/-] adult primary myogenic cells than it is on WT. (**B**) Mutations were made in the EphA7 and ephrin-A5 coding sequences of C2C12 myoblasts using double-nickase targeting, and single clones were expanded and analyzed by RT-PCR for inactivating mutations. Note: The red box in B indicates provisional sequencing of the targeted mutation in EphA7. (**C**) In high serum conditions, differentiation (indicated by expression of myogenin) is minimal in low density cultures of unedited, EphA7[KO], and ephrin-A5[KO] C2C12 cells. Unedited C2C12 cells respond to high density by differentiating even in the presence of high serum, but EphA7[KO] and ephrin-A5[KO] C2C12 cells do not. At either low or high densities, unedited and EphA7[KO] C2C12 cells differentiate in high serum in response to exposure to EphA7-Fc, but ephrin-A5[KO] C2C12 cells do not. Table shows all significance comparisons by density, substrate, and genotype. (**D**) Unedited and EphA7[KO] C2C12 cells phosphorylate ERK1/2 following 10′ exposure to soluble EphA7-Fc to a

*Figure 8 continued on next page*

*Figure 8 continued*

greater degree than do ephrin-A5$^{KO}$ C2C12 cells; all values were normalized to loading control. Error bars = SEM, p $\leq$ 0.01 (**), 0.001 (***) or 0.0001 (****). Scale bars = 50 μm.

The online version of this article includes the following source data for figure 8:

**Source data 1.** Expression of myogenin in unedited, EphA7KO, and ephrin-A5KO cells at high and low density, with and without exogenous EphA7-Fc.

phenocopy primary myoblasts from *EphA7*$^{-/-}$ mice with lower levels of myogenin expression, fewer, smaller myotubes with fewer nuclei, and sustained proliferation even at high density. We compared unedited, EphA7$^{KO}$, and ephrin-A5$^{KO}$ C2C12 cells grown in high serum at either high or low density, in the presence or absence of EphA7-Fc (*Figure 8C*). While all three lines showed minimal expression of myogenin (indicating a lack of differentiation) at low density on an unmodified laminin substrate, unedited C2C12 cells increased their differentiation when grown at high density as would be predicted by the community effect, while EphA7$^{KO}$ and ephrin-A5$^{KO}$ cells did not. In contrast, modifying the substrate with EphA7-Fc led to increased differentiation in both unedited and EphA7$^{KO}$ cells, even at low density and high serum, while differentiation in ephrin-A5$^{KO}$ cells was not rescued. When high density was combined with exposure to EphA7-Fc, differentiation by EphA7$^{KO}$ and ephrin-A5$^{KO}$ cells was not significantly different from their response to EphA7-Fc at low densities, while unedited C2C12 cells showed even more enhanced differentiation.

Reverse signaling by EphAs is less common than by EphBs, as ephrin-As are GPI-linked and thus lack an intracellular domain which would directly transduce the signal. However, in the case of EphA7:ephrin-A5 interactions, other groups have identified reverse signaling interactions that affect cell migration, adhesion, and lineage commitment (*Konstantinova et al., 2007*; *Haustead et al., 2008*; *Wimmer-Kleikamp et al., 2008*). One mechanism by which ephrin-A5 can transduce the EphA7 signal is via recruitment of Fyn, a nonreceptor tyrosine kinase that activates downstream effectors such as Src, ERK, Rac, AKT, paxillin and p75; in other cellular contexts this interaction regulates integrin-mediated signaling and adhesion, cell growth and survival, cytoskeletal remodeling, and cell motility (*Holen et al., 2008*; *Lim et al., 2008*). To test whether intracellular signaling following EphA7 stimulation is affected in ephrin-A5$^{KO}$ cells, we assayed activation of a potential downstream pathway by quantifying ERK1/2 phosphorylation in unedited, EphA7$^{KO}$, and ephrin-A5$^{KO}$ C2C12 cells in response to EphA7-Fc. We found that unedited and EphA7$^{KO}$ C2C12 cells rapidly and robustly phosphorylated ERK1/2, but ephrin-A5$^{KO}$ C2C12 cells have significantly decreased phosphorylation after 10' of exposure to EphA7-Fc (*Figure 8D*). Based on these data, we conclude that ephrin-A5 is a strong candidate for the EphA7 receptor on myoblasts.

Taken together, these results support a role for EphA7:ephrin-A5 interactions in induction of terminal differentiation during skeletal muscle development and regeneration, and provide a novel molecular mechanism for the community effect. While several other interactions mediating this effect have previously been described, we believe this study is the first to identify a heterotypic, non-cell-autonomous signal between myogenic cells at distinct stages of differentiation. As such, it provides a mechanism for rapidly amplifying the number of differentiated cells in a population once a sufficient local density has been reached and potential fusion partners are readily available. Further, the observation that cells lacking ephrin-A5 show an even more pronounced failure to differentiate than those lacking EphA7 suggests the possibility that threshold levels of ephrin-A5 stimulation may be necessary to initiate even stochastic differentiation.

## Discussion

Skeletal muscle is unique among vertebrate somatic tissues in that skeletal myofibers, the functional unit of the tissue, are syncytial cells formed by fusion of hundreds or thousands of terminally-differentiated mononuclear cells (myocytes). As such, skeletal muscle is particularly amenable to the use of contact-mediated signaling pathways to induce and coordinate cellular activities such as commitment to differentiation (reviewed in *Krauss et al., 2017*). Indeed, cell-cell contact and adhesion-based signals have been shown to be strongly promyogenic even in the presence of soluble factors that promote myoblast proliferation and thereby inhibit myocyte differentiation. Adhesion proteins such as cadherins, their multifunctional coreceptors Cdo and Boc, and the netrin-3 receptor neogenin are necessary and sufficient to promote robust myogenesis in vitro and in vivo (reviewed in

*Krauss et al., 2005*). It is important to note that the enhanced myogenesis brought about by these juxtacrine interactions is distinct from promotion of myocyte fusion, which has recently been shown to require a class of proteins in vertebrate muscle that do not appear to be related to classical contact-mediated signaling (*Millay et al., 2013*; *Quinn et al., 2017*; *Shi et al., 2017*; *Zhang et al., 2017*), and instead appears to derive from upregulation of MyoD-dependent transcription (reviewed in *Krauss et al., 2017*).

In this work we present data suggesting that contact-mediated signaling interactions between EphA7 and ephrin-A5 promote collective myogenic differentiation during mammalian muscle development and regeneration. EphA7 is one of a large family of receptor tyrosine kinases that are best characterized as promoting repulsion during cell migration and cell sorting at domain boundaries (reviewed in *Ventrella et al., 2017*). However, interactions between Ephs and their ephrin ligands have also been shown to directly affect differentiation: for example, EphB2:ephrinB1 interactions within the osteoblast lineage promote osteoblast differentiation by stimulating nuclear translocation of the transcriptional coactivator TAZ, and in multiple neuronal and stem cell lineages Eph/ephrin signaling modulates cell fate choice both directly and indirectly (reviewed in *Wilkinson, 2014*). EphA7 is a particularly likely candidate for a pro-differentiation molecule because it has been characterized as a tumor suppressor in several cancers (*Wang et al., 2005*; *Nakanishi et al., 2007*; *Oricchio et al., 2011*) as well as a negative regulator of proliferation through reverse signaling with ephrin-A2 in the adult stem cell niches of the hippocampus (*Holmberg et al., 2005*) and hair follicle (*Genander et al., 2010*). Three different splice variants of EPHA7 have been identified, encoding one full length (FL) and two truncated (T1 and T2) proteins lacking the catalytic kinase domain (*Holmberg et al., 2000*), as have two splice variants of ephrin-A5 (one lacking an alternate exon) (*Lai et al., 1999*). All three EphA7 proteins can signal in reverse via ephrin-A5, and in some contexts such as the closing neural tube the switch from full-length to truncated (T1) EphA7 appears to mediate a switch from repulsive to adhesive interactions (*Holmberg et al., 2000*); differential roles for full-length vs. truncated ephrin-A5 are less well described (*Li et al., 2001*). In the context of skeletal muscle regeneration, we detected mRNA expression of all three variants of EphA7 and both variants of ephrin-A5 (*Figure 1—figure supplement 1*); future work will explore the potential heterogeneity within the population and its mechanistic implications during myogenesis.

Based on our data, we propose a model in which EphA7 non-cell-autonomously promotes rapid and coordinated differentiation in populations of myogenic cells in a density-dependent manner (*Figure 9A*). An initially small number of ephrin-A5$^+$, EphA7$^-$ myoblasts will experience a combination of intrinsic and extrinsic factors that induce them to commit to differentiation, exit the cell cycle, and upregulate myogenin (stochastic commitment). Once this has occurred, these newly-differentiated myocytes will upregulate expression of EphA7, which will signal to adjacent ephrin-A5$^+$ myoblasts via cell-cell contact and promote their differentiation (synergistic commitment). This rapidly differentiating population of adjacent cells can subsequently fuse into myotubes/myofibers to form the functional units of skeletal muscle. In the absence of EphA7, while stochastic commitment still occurs, synergistic commitment via the community effect is lost, leading to decreased numbers of myocytes available for fusion (*Figure 9B*). Thus, both developmental and regenerative myogenesis still occurs in muscle lacking EphA7, but it progresses more slowly and produces smaller myofibers containing fewer myonuclei. Future studies will identify the mechanism by which EphA7 stimulation is transduced in ephrin-A5 expressing myoblasts, and what the molecular consequences of this interaction are. Given that other molecules which promote specification to myogenesis by cell-cell contact act via upregulation of MyoD-dependent gene transcription, it is intriguing to speculate that upregulation of myogenin and commitment to terminal differentiation could be a direct consequence of EphA7 reverse signaling in myoblasts.

## Materials and methods

### Muscle satellite cell isolation and culture

All experiments involving mice were conducted in accordance with National Institutes of Health and University of Missouri Institutional Animal Care and Use Committee guidelines. Adult mouse myoblasts were isolated by our published methods (*Capkovic et al., 2008*). Briefly, mice were euthanized, hindlimbs removed and skinned, and muscles removed in Dulbecco's PBS (DPBS). Following

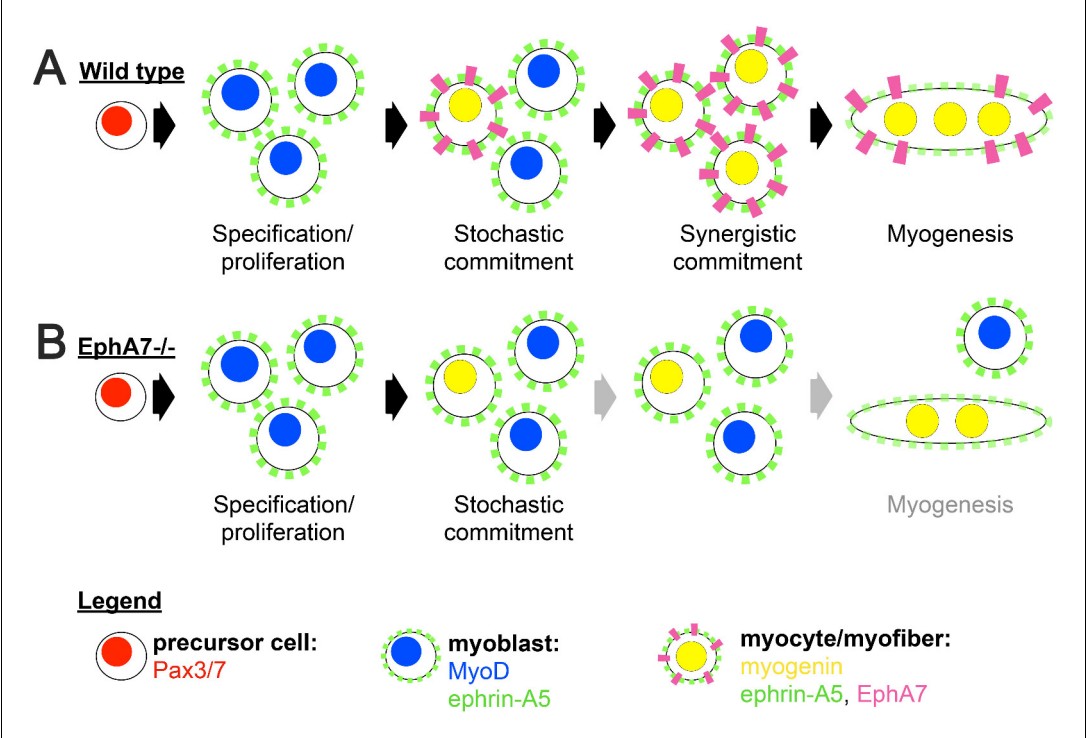

**Figure 9.** Proposed model. (**A**) We propose a model in which initial, stochastic commitment of individual myoblasts results in their expression of EphA7, which then induces rapid, synergistic commitment of adjacent ephrin-A5-expressing myoblasts via the community effect. (**B**) In the absence of EphA7, stochastic commitment still occurs but synergistic commitment does not, resulting in a decreased number of myocytes and later in fewer, smaller syncytial myofibers.

physical and enzymatic dissociation, cell slurries were filtered and pelleted then plated in Ham's F12 (Invitrogen) supplemented with 15% horse serum (Equitech), 5 nM FGF2, and either penicillin/streptomycin (Sigma) or puromycin (Sigma) on gelatin-coated plates. Cells were maintained at 37°C and 5% $CO_2$ in a humidified incubator. For differentiation, cells were washed briefly with cold DPBS and switched to Kaighn's F-12 supplemented with 2% horse serum and penicillin/streptomycin or puromycin.

Viable myofiber explants were isolated from EDL muscles using our published methods (*Cornelison and Wold, 1997*; *Cornelison et al., 2004*). As above, muscles were dissected, but were not physically dissociated. After collagenase digestion, free-floating myofibers were picked with a glass pipette and transferred into growth medium for culture as above.

## EphA7-Fc functionalized coverslips

EphA7-Fc functionalized coverslips, together with control laminin-coated coverslips, were freshly prepared the day of seeding. HCl-washed and aminopropyltriethoxysilane (APTES, Acros Organics) treated coverslips were incubated for 3 hours at 37°C with 1–2 mg/mL laminin (Sigma) diluted 1:100 in DPBS. Laminin was then gently aspirated and coverslips were washed twice with DPBS. Half of the coverslips were then incubated for 1.5 hours at 37°C with a 5 µg/mL solution of EphA7-Fc in DPBS, whereas control ones were incubated with only DPBS. The coverslips were washed once with DPBS and placed in multi-well plates with medium for 30' prior to cell seeding.

## In vivo muscle injury

Mice were anesthetized with 1.2% Avertin IP, then 70 µL of sterile 1.2% $BaCl_2$ in water was injected into the left TA. Mice were allowed to recover on a warmed platform until the effects of anesthesia were no longer noticeable and experimental injury was confirmed by the presence of foot drop. Mice were housed separately for 3–28 days before the injured muscles were harvested for analysis.

## Antibody staining

For fluorescence immunohistochemistry of muscle sections, muscles were flash-frozen in liquid nitrogen-cooled isopentane and sectioned on a Leica cryostat. For fluorescence immunohistochemistry of mouse embryos, embryos were fixed overnight in 4% paraformaldehyde, washed in PBS, and equilibrated into 15% then 30% sucrose in PBS prior to freezing and sectioning.

For fluorescence immunocytochemistry of cultured cells, satellite cells prepared as described above were replated onto gelatin- or laminin-coated glass coverslips, allowed to adhere for 2 hr, then cultured for 12–96 hours and fixed in 4% ice-cold paraformaldehyde. Coverslips were blocked for 1 hour at room temperature with 10% normal goat serum containing 0.2% Triton X-100 then incubated with primary antibody diluted in 10% NGS overnight at 4°C. Coverslips were washed 3 × 10′ in PBS then incubated with secondary antibodies diluted in 10% NGS for 1 hour at RT, washed again, and mounted with Vectashield (Vector) with DAPI. For staining of adult myofiber-associated satellite cells, myofibers at the timepoints indicated in the text were fixed in 4% PFA, washed in PBS, and arranged on slides for staining as above.

Concentrations of primary antibodies used were rabbit anti-EphA7 (Santa Cruz Biotechnology, Inc; Antibody Registry AB_2099673) at 1:200; rabbit anti-laminin (Sigma-Aldrich, Antibody Registry AB_477163) at 1:300; mouse anti-eMyHC (clone F1.652, Developmental Studies Hybridoma Bank, Antibody Registry AB_528358) at 1:50; rat anti-CD34 (clone RAM34, eBioscience, Antibody Registry AB_467211) at 1:200; chicken anti-syndecan-4 (*Cornelison et al., 2004*) at 1:1500; mouse anti-MyoD (NCL-MyoD1, Leica, Antibody Registry AB_563997) at 1:10; mouse anti-Pax7 (Developmental Studies Hybridoma Bank, Antibody Registry AB_528428) at 1:10; mouse anti-myogenin (F5D, Developmental Studies Hybridoma Bank, Antibody Registry AB_2146602) at 1:5; mouse anti-myosin heavy chain (MF20, Developmental Studies Hybridoma Bank, Antibody Registry AB_2147781) neat. Secondary antibodies were raised in goat and conjugated with Alexa fluorophores (Invitrogen), and used at 1:500.

All fluorescent images were acquired and processed on an Olympus BX61 upright microscope using Slidebook software (Intelligent Imaging Innovations) or ImageJ (*Schneider et al., 2012*). Digital background subtraction was used to remove signal less than or equal to levels present in control samples processed without primary antibody, and was applied equally to the entire field.

For all quantitative comparisons of expression by immunohistochemistry, at least three biological replicates per genotype per timepoint per stain were analyzed; for cells grown on coverslips or single myofibers > 10 fields/coverslip or >20 myofibers/animal were scored and summed.

## Morphometric analysis

Myofiber cross-sectional area was automatically calculated based on laminin immunohistochemistry and (in injured/regenerated samples) the presence of central nuclei using SMASH (*Smith and Barton, 2014*). A minimum of three animals/genotype/timepoint were analyzed.

## RT-PCR

Intron-spanning primers were written to amplify EphA7 and ephrin-A5 splice isoforms and MYH3 (eMyHC):

```
EphA7V1: 5′AACCGGGAACAGTGTACGTC3′ F; 5′TGGTGCCTGGAAATTTAAAATGA3′ R
EphA7V2: 5′TGAGAAAGATCAACGGGAAAGGA3′; 5′CTGACAGGTGCTCATTTGTTAC3′ R
EphA7V3: 5′TCTACTTTCATTGCACCAAAACCT3′ F; 5′GCCCATCGTGTTTCCTGAGA3′ R
EphrinA5V1: 5′AGCAACCCCAGATTCCAGAGG3′ F; 5′ATCGAAAACACGATCATGAACACC3′ R
EphrinA5V2: 5′TTGTGAGACCAACAAATGACAC3′ F; 5′CCGTTTGATTGGGAAGGGGA3′ R
MYH3: 5′CGTTTTGGACATTGCGGGTT3′ F; 5′ACCGTCCGCATCTGTTGTAG3′ R
```

RNA was isolated from TA muscle samples and 1.5 μg was used as template for reverse transcription with SuperScript IV RT (Invitrogen). 1/20 of each reaction was used as template for 30 cycles of PCR amplification (except MYH3, which had 25 cycles) with annealing at 55°.

## Double-nickase knockout cells

CRISPR Double Nickase Plasmid (m2) targeting mouse EphA7 (sc-420197-NIC-2) and ephrin-A5 (sc-420123-NIC-2) were purchased from Santa Cruz Biotechnologies. C2C12 cells were purchased from

ATCC and were tested as mycoplasma-free. Cells were transfected with 2 µg of plasmid using Invitrogen Neon transfection system (1050 V; 30 ms; two pulses). 1.5 µg/mL puromycin was added to transfected cells 24 hours after transfection and every 4 days for 15 days then removed for one week. GFP+, puromycin-resistant clones were individually isolated and expanded then screened for inactivating mutations phenotypically and by RT-PCR and Western blot.

## EphA7-Fc stimulation of cells and Western blot

Primary satellite cells or C2C12 cells were cultured to 80% confluency in 10 cm dishes, then starved overnight (DMEM + 0.1% gentamycin). EphA7-Fc was pre-clustered with anti-human secondary antibody by incubating 100 µg/mL EphA7-Fc (R&D Systems) with 20 µg/mL goat anti-human IgG (Invitrogen) in 0.1% BSA for 30' at room temperature. Cells were washed twice with DPBS then incubated for 10' at 37 °C with DMEM + 0.5% FBS supplemented with either anti-human IgG/BSA without EphA7-Fc (control cells) or with pre-clustered EphA7-Fc at a final concentration of 5 µg/mL (stimulated cells). Cells were gently washed with cold DPBS and lysed in Pierce Protease+Phosphatase Inhibitor cocktail (Thermo Scientific) for 30' on ice, then scraped and soluble proteins were isolated by centrifugation. 20 µg of each lysate was separated by SDS-PAGE then transferred to PVDF membrane. Membranes were incubated in StartingBlock TBS (Thermo Scientific) for 2 hours at room temperature then incubated with mouse anti-phospho-ERK1/2 (Sigma; Antibody Registry AB_260729) at 1:2000 overnight. Membranes were washed 3 × 10' at room temperature with TBS + 0.05% Tween-20 (TBST) then incubated with anti-mouse HRP secondary antibody at 1:20,000 for 1 hour at room temperature. Membranes were washed 3 × 20' at room temperature with TBST then detected with SuperSignal West Femto Maximum Sensitivity Substrate (Thermo Scientific) and a UVP Camera System. Images were processed with Fiji image processing software. To compare protein loading, membranes were stripped with ReSTORE Plus (Thermo Scientific) then incubated with anti-calnexin (AbCam; Antibody Registry AB_2069006) at 1:2000 as above, except for detection with anti-rabbit HRP secondary antibody at 1:50,000.

## Statistical methods

Where indicated, samples were compared in Graphpad Prism by one-way ANOVA (Kruskal-Willis test) followed by Dunn's multiple comparison test or Wilcoxson Mann-Whitney test.

# Acknowledgements

This work was supported by the National Institutes of Health (grant number AR067450) to DDWC. The authors are very grateful to Dr. Fabio Rossi for providing RNA-seq data on EphA7 in FAPs, and to Dr. Tom Cheung, Dr. Bradley Olwin, and Dr. Benjamin Cosgrove for querying single cell RNA-seq data in muscle resident cell types.

# Additional information

### Funding

| Funder | Grant reference number | Author |
| --- | --- | --- |
| National Institutes of Health | AR067450 | DDW Cornelison |

The funders had no role in study design, data collection and interpretation, or the decision to submit the work for publication.

### Author contributions

Laura L Arnold, Alessandra Cecchini, Danny A Stark, Conceptualization, Formal analysis, Validation, Investigation, Visualization, Methodology; Jacqueline Ihnat, Rebecca N Craigg, Amory Carter, Sammy Zino, Investigation; DDW Cornelison, Conceptualization, Resources, Formal analysis, Supervision, Funding acquisition, Validation, Investigation, Visualization, Methodology, Project administration

## Author ORCIDs

DDW Cornelison https://orcid.org/0000-0003-4287-7524

## Ethics

Animal experimentation: All experiments involving mice were conducted in accordance with National Institutes of Health and University of Missouri Institutional Animal Care and Use Committee guidelines. All animals were handled in accordance with an approved Animal Care and Use Committee-approved protocol (#8325 to D Cornelison). All experiments were animals were designed in accordance with the '3 R's' and every effort was made to minimize pain and suffering.

## Decision letter and Author response

Decision letter https://doi.org/10.7554/eLife.53689.sa1
Author response https://doi.org/10.7554/eLife.53689.sa2

# Additional files

## Supplementary files

• Transparent reporting form

## Data availability

All data generated or analysed during this study are included in the manuscript and supporting files.

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
