## [Decision Letter]

**Acceptance summary:**

In this manuscript the authors provide compelling evidence that EphA7-EphinA5 partnership regulates density-dependent myogenic differentiation. The work is carefully executed and appropriately interpreted. The results provide a mechanistic understanding for the 'community effect'. The findings of this work will have implications for biologists across multiple fields.

**Decision letter after peer review:**

Thank you for sending your article entitled "EphA7 promotes myogenic differentiation via cell-cell contact" for peer review at *eLife*. Your article has been evaluated by three peer reviewers, including Andrew Brack as the Reviewing Editor and Reviewer #1, and the evaluation has been overseen by Marianne Bronner as the Senior Editor.

The reviewers have discussed the reviews with one another and the Reviewing Editor has drafted this decision to help you prepare a revised submission. The full reviews are included below for further information.

Summary:

In this manuscript Arnold et al. report on the essential contribution that EphA7 expression on nascent myofibers makes to the completion of skeletal myogenesis during muscle regeneration. The data fairly support the author's model, in which EphA7 expression on the surface of differentiation-committed myogenic progenitors and adjacent myocytes provides a key signal that promotes formation of mature myofibers during muscle regeneration.

The authors propose that localized foci of accelerated myogenesis occurs within high-density of EphA7 expressing cells, according to a stochastic model of progressive spreading of foci of skeletal myogenesis through the "so-called" community effect. EphA7-null mice had lower numbers of myofibers, which were smaller in area. Regeneration was compromised/delayed in these mice. Mutation of EphA7 in mice resulted in many fewer myogenin+ cells in vitro, without affecting the percentage of MyoD+ cells. Plating EphA7-null satellite cells on an EphA7-Fc substrate rescued myogenin expression, showing that EphA7 acts in this system as a ligand, rather than a receptor (in a reverse signaling fashion). Overall, the data shown in this manuscript provide novel insights into a largely under-investigated, yet very interesting topic of investigation.

The work presented appears carefully done and the figures are of high quality. There is a lack of experimentation to explain the mechanism of how EphA7 promotes differentiation. However, identification of EphA7 as a participant in a "community effect" is interesting. The following comments should be addressed.

Essential revisions:

Confounding variable when using germline deletion of EhrinA7:

1) The delayed regeneration process observed in EphA7 null mice suggests incomplete skeletal myogenesis in the absence of EphA7. However, given that the studies are performed in a general KO mouse model, it is unclear whether this is a cell autonomous defect of myogenic cells, or it derives from the lack of EphA7 expression in other cell types that might contribute to regeneration. These are the only in vivo data in the manuscript and therefore a key component of the manuscript.

Below we provide some possible solutions to address this limitation:

– Figure 5 shows a time course of regeneration and concludes that in the absence of EphA7, eMyHC expression persists in unfused myocytes or mononucleated fibers. A problem here is that mutation used is global, and the mice begin with only ~40% the number of satellite cells that WT mice have. It is not clear how much of the phenotype might be due to starting with fewer satellite cells vs. compromised/delayed differentiation. One way to address this might be to quantify eMyHC^+^ nascent myofibers at earlier time points (5 dpi). The figure suggests that things may have "caught up" to a reasonable extent at that time point, only to be deficient again at later steps. More difficult, but perhaps more rigorous would be to use the Pax7CreER driver with the DTA system to kill off satellite cells, but only ~60% of them, so that an EphA7^+^ mouse with a similar deficit in satellite cell numbers could be compared with EphA7-null mice.

– The central point is that extracellular EphA7 functions as a ligand to promote differentiation via cell to cell contact. Would a transplantation assay help? For example, transplanting EphA7 null SCs into EphA7 null muscle (increase SC number but no benefit of fusion) versus WT into null. Differentiation/fusion would be scored based on donor cell contribution.

– Although the optimal strategy would consist in the generation and analysis of a mouse model of cell-specific conditional genetic deletion of EphA7, or experiments of cell transplantation (e.g. EphA7 deficient satellite cells into WT muscles subjected to injury), this reviewer understands that these experiments would require a substantial amount of time. The authors could verify by qPCR that EphA7is only (or most largely) expressed in satellite cells (committed to differentiation) and nascent myofibers, but not in other cell types – e.g. macrophages, endothelial, FAPs – isolated from regenerating muscles at different time points post-injury. I also note that equivalent information could be retrieved by publicly available RNAseq datasets on muscle-derived cells isolated during muscle regeneration. Overall, providing definitive evidence that EphA7 expression is mostly confined to myoblasts and myofibers within regenerating muscles, as also suggested by immunofluorescence analysis in muscle sections (Figure 1), could in my opinion wave the request of generating mice with cell-specific conditional genetic deletion of EphA7.

Further characterization of Eph and Ehprin expression:

2) Addition of extracellular domain induces differentiation to WT and KO, suggesting EphA7 must bind to another receptor. What is the Ephin is expressed in myoblasts? Are there other EphA and Ephrin proteins expressed in myocytes and myoblasts.

3) Eph receptors are probably best known for acting in cell repulsion events in CNS development. In the Discussion, it is mentioned that one of the truncated splice variants of EphA7 promotes cell adhesion, which would better fit its role in myogenic differentiation. It should be straightforward to determine which isoforms are present in differentiating satellite cell cultures, helping to explain what is presumably an adhesion, not repellent, function.

Further characterization of Ephrin A7 molecular function:

4) Does ectodomain regulate differentiation in clonal or low-density cultures (avoid cell to cell contact). It shouldn't if the authors model is correct.

5) Does the ectodomain act as a decoy receptor? Measure RTK signaling in WT, null and Ectodomain-Fc cultures.

6) Does EphrinA7 act as a soluble ligand? Add ectodomain to media or in Boyden chamber and examine phenotypes.

Need for further quantification of data:

7) Figure 1 – needs quantification and statistical analysis of% of eMyHC positive fibers that also express EphA7 expression, and vice versa, at the different timepoints. Given that expression of EphA7 seems mostly confined to nascent fibers, it would be also informative to perform a quantification of eMyHC and EphA7 by PCR in the whole muscles at each of the timepoints. This analysis will also inform on whether the regulation of EphA7 expression during muscle regeneration occurs at the transcriptional or post-transcriptional level.

8) Still in Figure 1, the authors should perform immunofluorescence analysis of EphA7 expression in satellite cells, as there is clearly a laminin-associated green staining (EphA7) observed in muscle sections at 5 dpi that could derived from satellite cells. Co-staining with MyoD, Myogenin or Pax7 should be performed to fully address this issue. This data should also corroborate conclusions from Figure 2.

9) Figure 5D needs statistical analysis. Also, it is stated that, "we find that muscle myofiber hypertrophy is delayed in *EphA7*^-/-^ muscles, even after taking into account the original deficit in fiber size". An additional graph and statistics should compare uninjured and 28 dpi EphA7-null fiber sizes to provide evidence for this conclusion.

Reviewer #1:

In this manuscript, Arnold et al. demonstrate a role for the EphA7 in muscle differentiation.

The authors use a combination of in vivo mouse genetics, expression analysis and cell culture to argue that the EphA7 is expressed in myocytes to promote differentiation of neighboring myoblasts. functions as a ligand. Experiments are performed to a high standard. The authors conclusions are consistent with the data.

I have a few points that are important to address:

Addition of extracellular domain induces differentiation to WT and KO. Therefore it must bind to another receptor.

What is the Ephin is expressed in myoblasts?

Are there other EphA and Ephrin proteins expressed in myocytes and myoblasts?

Does ectodomain regulate differentiation in clonal or low density cultures (avoid cell to cell contact)? It shouldn't if the authors model is correct.

Does the ectodomain act as a decoy receptor? Measure RTK signaling in WT, null and Ectodmain-Fc cultures?

Does EphrinA7 act as a soluble ligand? Add ectodomain to media or in Boyden chamber and examine phenotypes.

Reviewer #2:

In this manuscript Arnold et al. report on the essential contribution that EphA7 expression on nascent myofibers makes to the completion of skeletal myogenesis during muscle regeneration. The data fairly support the author's model, in which EphA7 expression on the surface of differentiation-committed myogenic progenitors and adjacent myocytes provides a key signal that promotes formation of mature myofibers during muscle regeneration.

The authors propose that localized foci of accelerated myogenesis occurs within high-density of EphA7 expressing cells, according to a stochastic model of progressive spreading of foci of skeletal myogenesis through the "so-called" community effect.

Overall, the data shown in this manuscript provide novel mechanistic insights into a largely under-investigated, yet very interesting topic of investigation.

Some additional piece of experimental evidence in support to the proposed model would further strength this manuscript, as indicated in the specific points – see below.

1) Figure 1 – needs quantification and statistical analysis of% of eMyHC positive fibers that also express EphA7 expression, and vice versa, at the different timepoints. Given that expression of EphA7 seems mostly confined to nascent fibers, it would be also informative to perform a quantification of eMyHC and EphA7 by PCR in the whole muscles at each of the timepoints. This analysis will also inform on whether the regulation of EphA7 expression during muscle regeneration occurs at the transcriptional or post-transcriptional level.

2) Still in Figure 1, the authors should perform immunofluorescence analysis of EphA7 expression in satellite cells, as there is clearly a laminin-associated green staining (EphA7) observed in muscle sections at 5 dpi that could derived from satellite cells. Co-staining with MyoD, Myogenin or Pax7 should be performed to fully address this issue. This data should also corroborate conclusions from Figure 2.

3) The delayed regeneration process observed in EphA7 null mice suggests incomplete skeletal myogenesis in the absence of EphA7. However, given that the studies are performed in a general KO mouse model, it is unclear whether this is a cell autonomous defect of myogenic cells, or it derives from the lack of EphA7 expression in other cell types that might contribute to regeneration. Although the optimal strategy would consist in the generation and analysis of a mouse model of cell-specific conditional genetic deletion of EphA7, or experiments of cell transplantation (e.g. EphA7 deficient satellite cells into WT muscles subjected to injury), this reviewer understands that these experiments would require a substantial amount of time. I rather suggest that the authors verify by qPCR that EphA7is only (or most largely) expressed in satellite cells (committed to differentiation) and nascent myofibers, but not in other cell types – es macrophages, endothelial, FAPs – isolated from regenerating muscles at different time points post-injury. I also note that equivalent information could be retrieved by publicly available RNAseq datasets on muscle-derived cells isolated during muscle regeneration. Overall, providing definitive evidence that EphA7 expression is mostly confined to myoblasts and myofibers within regenerating muscles, as also suggested by immunofluorescence analysis in muscle sections (Figure 1), would in my opinion wave the request of generating mice with cell-specific conditional genetic deletion of EphA7.

4) Figure 7B. By close inspection of the pictures it seems to this reviewer that satellite cell cultures from EphA7 null mice display evident morphological differences, as compared to WT cultures, regardless of the presence of EphA7-Fc. In particular, there are several phalloidin-positive flat cells showing a phenotype reminiscent of senescent or endothelial like cells. The authors should better analyze and explain these morphological discrepancies, and understand whether they are due to FACS contamination, cultural artifacts or a reproducible acquisition of lineage heterogeneity by satellite cells, caused by EphA7 deficiency.

5) Figure 8. The proposed model is not of "immediate catch" by first glance. Furthermore, the understanding of this figure is limited by lack of color identity information for nuclear purple and nuclear green. I suggest that the authors substantially revise this figure to make their proposed model clearer.

Reviewer #3:

This manuscript reports on the role of EphA7 in differentiation of skeletal muscle precursor cells. This is related to the "community effect" first described by Gurdon, an interesting phenomenon for which few molecular regulators have been identified. The authors carefully document the expression pattern of EphA7 in developing and regenerating skeletal muscle, and also in satellite cells on isolated myofibers and in primary culture in vitro. EphA7 is induced in differentiating cells, correlating temporally with myogenin and MyHC expression. EphA7-null mice had lower numbers of myofibers, which were smaller in area. Regeneration was compromised/delayed in these mice. Mutation of EphA7 in mice resulted in many fewer myogenin+ cells in vitro, without affecting the percentage of MyoD+ cells. Plating EphA7-null satellite cells on an EphA7-Fc substrate rescued myogenin expression, showing that EphA7 acts in this system as a ligand, rather than a receptor (in a reverse signaling fashion).

The work presented appears carefully done and the figures are of high quality. There is a lack of experimentation to explain mechanisms of how EphA7 promotes differentiation or how its expression is induced, which may detract from publication in a journal like *eLife*. However, identification of EphA7 as a participant in a "community effect" is interesting. The following comments should be addressed.

1) Figure 5 shows a time course of regeneration and concludes that in the absence of EphA7, eMyHC expression persists in unfused myocytes or mononucleated fibers. A problem here is that mutation used is global, and the mice begin with only ~40% the number of satellite cells that WT mice have. It is not clear how much of the phenotype might be due to starting with fewer satellite cells vs. compromised/delayed differentiation. One way to address this might be to quantify eMyHC^+^ nascent myofibers at earlier time points (5 dpi). The figure suggests that things may have "caught up" to a reasonable extent at that time point, only to be deficient again at later steps. More difficult, but perhaps more rigorous would be to use the Pax7CreER driver with the DTA system to kill off satellite cells, but only ~60% of them, so that an EphA7^+^ mouse with a similar deficit in satellite cell numbers could be compared with EphA7-null mice.

2) Figure 5D needs statistical analysis. Also, it is stated that, "we find that muscle myofiber hypertrophy is delayed in *EphA7*^-/-^ muscles, even after taking into account the original deficit in fiber size". An additional graph and statistics should compare uninjured and 28 dpi EphA7-null fiber sizes to provide evidence for this conclusion.

3) Eph receptors are probably best known for acting in cell repulsion events in CNS development. In the Discussion, it is mentioned that one of the truncated splice variants of EphA7 promotes cell adhesion, which would better fit its role in myogenic differentiation. It should be straightforward to determine which isoforms are present in differentiating satellite cell cultures, helping to explain what is presumably an adhesion, not repellent, function.

---

## [Author Response]

Essential revisions:Confounding variable when using germline deletion of EphrinA7:1) The delayed regeneration process observed in EphA7 null mice suggests incomplete skeletal myogenesis in the absence of EphA7. However, given that the studies are performed in a general KO mouse model, it is unclear whether this is a cell autonomous defect of myogenic cells, or it derives from the lack of EphA7 expression in other cell types that might contribute to regeneration. These are the only in vivo data in the manuscript and therefore a key component of the manuscript.Below we provide some possible solutions to address this limitation:– Figure 5 shows a time course of regeneration and concludes that in the absence of EphA7, eMyHC expression persists in unfused myocytes or mononucleated fibers. A problem here is that mutation used is global, and the mice begin with only ~40% the number of satellite cells that WT mice have. It is not clear how much of the phenotype might be due to starting with fewer satellite cells vs. compromised/delayed differentiation. One way to address this might be to quantify eMyHC^+^ nascent myofibers at earlier time points (5 dpi). The figure suggests that things may have "caught up" to a reasonable extent at that time point, only to be deficient again at later steps. More difficult, but perhaps more rigorous would be to use the Pax7CreER driver with the DTA system to kill off satellite cells, but only ~60% of them, so that an EphA7^+^ mouse with a similar deficit in satellite cell numbers could be compared with EphA7-null mice.– The central point is that extracellular EphA7 functions as a ligand to promote differentiation via cell to cell contact. Would a transplantation assay help? For example, transplanting EphA7 null SCs into EphA7 null muscle (increase SC number but no benefit of fusion) versus WT into null. Differentiation/fusion would be scored based on donor cell contribution.– Although the optimal strategy would consist in the generation and analysis of a mouse model of cell-specific conditional genetic deletion of EphA7, or experiments of cell transplantation (e.g. EphA7 deficient satellite cells into WT muscles subjected to injury), this reviewer understands that these experiments would require a substantial amount of time. The authors could verify by qPCR that EphA7is only (or most largely) expressed in satellite cells (committed to differentiation) and nascent myofibers, but not in other cell types – e.g. macrophages, endothelial, FAPs – isolated from regenerating muscles at different time points post-injury. I also note that equivalent information could be retrieved by publicly available RNAseq datasets on muscle-derived cells isolated during muscle regeneration. Overall, providing definitive evidence that EphA7 expression is mostly confined to myoblasts and myofibers within regenerating muscles, as also suggested by immunofluorescence analysis in muscle sections (Figure 1), could in my opinion wave the request of generating mice with cell-specific conditional genetic deletion of EphA7.Further characterization of Eph and Ephrin expression:2) Addition of extracellular domain induces differentiation to WT and KO, suggesting EphA7 must bind to another receptor. What is the Ephin is expressed in myoblasts? Are there other EphA and Ephrin proteins expressed in myocytes and myoblasts.3) Eph receptors are probably best known for acting in cell repulsion events in CNS development. In the Discussion, it is mentioned that one of the truncated splice variants of EphA7 promotes cell adhesion, which would better fit its role in myogenic differentiation. It should be straightforward to determine which isoforms are present in differentiating satellite cell cultures, helping to explain what is presumably an adhesion, not repellent, function.Further characterization of Eph A7 molecular function:4) Does ectodomain regulate differentiation in clonal or low-density cultures (avoid cell to cell contact). It shouldn't if the authors model is correct.5) Does the ectodomain act as a decoy receptor? Measure RTK signaling in WT, null and Ectodomain-Fc cultures.6) Does EphA7 act as a soluble ligand? Add ectodomain to media or in Boyden chamber and examine phenotypes.Need for further quantification of data:7) Figure 1 – needs quantification and statistical analysis of% of eMyHC positive fibers that also express EphA7 expression, and vice versa, at the different timepoints. Given that expression of EphA7 seems mostly confined to nascent fibers, it would be also informative to perform a quantification of eMyHC and EphA7 by PCR in the whole muscles at each of the timepoints. This analysis will also inform on whether the regulation of EphA7 expression during muscle regeneration occurs at the transcriptional or post-transcriptional level.8) Still in Figure 1, the authors should perform immunofluorescence analysis of EphA7 expression in satellite cells, as there is clearly a laminin-associated green staining (EphA7) observed in muscle sections at 5 dpi that could derived from satellite cells. Co-staining with MyoD, Myogenin or Pax7 should be performed to fully address this issue. This data should also corroborate conclusions from Figure 2.9) Figure 5D needs statistical analysis. Also, it is stated that, "we find that muscle myofiber hypertrophy is delayed in *EphA7*^-/-^ muscles, even after taking into account the original deficit in fiber size". An additional graph and statistics should compare uninjured and 28 dpi EphA7-null fiber sizes to provide evidence for this conclusion.

We have edited the manuscript and figures as well as generated new data and figures, and we thank the reviewers for the significant improvements that their suggestions have made in our manuscript. The most significant change is a new figure (Figure 8) in which we show that ephrin-A5 meets the criteria for the EphA7 receptor in this system, thus providing a more complete mechanism than we had in the previous version; all major revisions are described below.

We have added a new panel to Figure 1 (Figure 1B) in which the co-expression of EphA7 and eMyHC across the timecourse is quantified. As noted in the figure legend, at later timepoints a small fraction of laminin-bounded fibers is scored positive for EphA7 but negative for eMyHC due to the thresholds determined during imaging (normalized to a no-primary control in each case). However, we believe that in actuality there are no EphA7^+^/eMyHC- fibers. We present the data as we do (and with this caveat) because it seems the best and most honest way to approach it, however if the reviewers have suggestions regarding either the data presentation or the text/figure legend we would be grateful for advice. As suggested, we provide an RT-PCR timecourse for all three splice isoforms of EphA7 (and both splice isoforms of ephrin-A5, see below) as well as eMyHC in a supplementary figure (Figure 1—figure supplement 1). We do not draw any mechanistic conclusions from these data in this manuscript because that would require not only protein/RNA resolution at the level of individual cells in situ but downstream signaling analyses; however, we thank the reviewers for suggesting the experiment and plan to follow up on these intriguing results if possible.

We have included a new panel in Figure 5 (G) showing average fiber size in WT and *EphA7*^-/-^ regenerating muscle at each timepoint (in addition to the binned areas shown previously), with statistics comparing genotypes at each timepoint.

We remade Figure 7B, to remove potential confusion regarding cell morphology and identity. We have also included additional text to clarify that except for a higher representation of proliferating cells, there is no obvious difference in the appearance of satellite cell cultures from WT and *EphA7*^-/-^ mice.

To provide a molecular mechanism for the activity of EphA7, we have included new data (now Figure 8) and accompanying text supporting ephrin-A5 as the EphA7 receptor on myoblasts. This figure includes data showing that ephrin-A5 protein expression is increased in *EphA7*^-/-^ primary satellite cells; that in C2C12 cells high density promotes differentiation in unedited but not EphA7^KO^ or ephrin-A5^KO^ cells, while EphA7 ectodomain promotes differentiation in unedited and EphA7^KO^ at either high or low densities but never in ephrin-A5^KO^ cells; and that exposure to EphA7 ectodomain rapidly induces ERK1/2 phosphorylation in unedited and EphA7^KO^ cells which is more than twofold reduced in ephrin-A5^KO^ cells. We are very pleased with this figure and hope that it will be useful to clarify the roles of density, EphA7 as a pro-differentiation stimulus, and ephrin-A5 as the receptor for that non-cell-autonomous stimulus. Due to the Covid-19 lockdown on the MU campus, we were unable to complete the confirmation of the sequence deleted in the EphA7^KO^ line of C2C12s; this uncertainty is indicated graphically on the figure itself.

We remade the model (now Figure 9) to clarify and to include new data. While co-expression of Pax7 and MyoD, and of MyoD and myogenin, are important transient states in myogenesis, knowing this is not necessary to understand the model. We have therefore removed the purple (red + blue) and green (yellow + blue) nuclei to make the model simpler. In addition, since our data now include ephrin-A5 as the EphA7 receptor on myoblasts, we have included this expression in the schematic although it does not change over the course of differentiation.

To address the concern that EphA7 may be expressed in nonmuscle cell types and that this might be responsible for at least some of the observed phenotype, we have consulted with Dr. Fabio Rossi who has generated bulk RNA-seq datasets for the most likely nonmuscle cell type to influence muscle differentiation (FAPs), as well as with Dr. Tom Cheung, Dr. Bradley Olwin, and Dr. Benjamin Cosgrove, who have each independently generated single cell RNA-seq datasets covering all resident cell types. Dr. Rossi confirms that EphA7 is not expressed by FAPs and we note this in the text. However, neither EphhA7 nor ephrin-A5 RNA are detectable in any cell types during muscle regeneration in any of the single-cell datasets we consulted, making interpretation of those data difficult. While we are disappointed that the best we can offer is negative data, based our primary myogenic cell cultures and C2C12 knockout experiments we remain confident that the effect we observe in vivo is primarily or entirely due to myocyte-myoblast interactions. Unfortunately, definitive proof for this will not be available for some time (we are acquiring the ephrin-A5 conditional knockout line from a collaborator as frozen sperm).

We have expanded the text to more explicitly describe the non-cell-autonomous effect we propose, and to note that exposure to the EphA7-Fc would therefore be expected to act as a proxy for high cell density. The data in new Figure 8 are also presented in a way designed to emphasize this point.